# OpenReview forum: "GraphShield: Graph-Theoretic Modeling of Network-Level Dynamics for Robust Jailbreak Detection"
_ICLR.cc/2026/Conference — ICLR 2026 Poster_

### Official Review · Reviewer_fKnw · 2025-10-24

**Soundness:** 3
**Presentation:** 2
**Contribution:** 3
**Rating:** 4
**Confidence:** 3

**Summary:**

This paper introduces a novel way of defending against jailbreaks using internal LLM signals, that don't just look at token-level signals (like linear probes), but also look at the flow of information. I think the idea is very interesting and nice. However, I have concerns about the exposition, framing, baselines, and strength of results. I lean towards rejection due to the concerns, but I expect I could change my mind.

**Strengths:**

The idea is very interesting and as far as I know, novel. My intuitive understanding (which could be wrong) is basically not looking at just a probe, but understanding more of a sense of how information is being used in the network. This is an interesting idea. I think it would be really great to have a paragraph in the introduction explaining this at a high level.

**Weaknesses:**

- **Improve the exposition and writing**. In general, I think the exposition of the paper could be greatly improved.
    - For a machine learning reader that is less familiar with neuroscience, I recommend framing more and adding more explanatory information. e.g., "jailbreak behavior in LLMs is best understood as an emergent property of information routing rather than an isolated token- or activation level phenomenon.". What does this mean, precisely? I understood this eventually, but adding figures in the introductoin would be helpful. I think Figure 1 is fairly good—adding it earlier would be good.
    - Introduction claims "prior attempts rely on surface level signals." I don't think this is quite true—in what way do classifiers rely on surface level signals? The concern there is they are "tied to training taxonomies"—what jailbreak defense isn't? This framing seems to be off to me. What is exactly the limitation of a training taxonomy? In fact, it can be helpful—e.g., Sharma et al. (2025) Constitutional Classifiers paper shows how a constitution can be used to train. I'd also suggest citing this paper.
    - On page 4, where the maths is introduced, I suggest adding more exposition and in general explaining what is happening.
    - Adding an extra figure on page 4 to explain the different maths terms would also help, and understand the sparsified graph. I think you have this graph later.
    - On page 5, explain the motivation behind different features for the graph, and ideally in some easier to understand way. I would guess that jailbreaks break the refusal edge connectivity, so that etiher the refusal never comes up, or it comes up and it it isn't routed to the output. Where is that captured here?
    - On page 7, adding some bold and some flow going through the results would improve how easily the results are understood.
- Judging procedure
    - I don't understand why you use a keyword pattern filter? I would prefer something stronger, like the rubric approach used in the StrongReject paper.
- Easier comparison
    - It would be easier to compare results in Table 1 if we fixed the thresholds to have a fixed benign refusal rate.
- Baseline
    - I'd like to see a baseline of a simple linear probe to see the benefit of using connectivity.
- I'm concerned about the strength of defences, given that decipher attacks and some adaptive attacks (without adaptive defences) seem to beat the method. In fact, given this, this suggests that these defences, as they are, might simply not be that robust. I think the issue lies with the use of anchor tokens that are pain english (see questions below). So I am concerned about this.

**Questions:**

- I want to understand use of anchor tokens—it seems like you're probing for indication of refusal. But I don't fully get it.
- And it's not clear whether the anchor tokens generalize e.g., across languages? I'd like to see a simple experiment where we try to break the use of anchor tokens by translating text into another language. Indeed, I think the poor performance of Cipher might be due to this (i.e., the relevant token similarities are very different, so we get poor generalization)
- Did you consider alternative approaches for anchor token? E.g., you could find refusal vectors at each layer (more like training a linear probe). This might help with the above issues.
- I'd love to understand the edge weights better. Are we looking at the edge weights layer to layer (i.e., understanding if information related to the anchor tokens progagates?)
- I don't follow why the ASR is so different for Llama-2 and Vicuna for e.g., Llama guard in Table 1? Llama guard is an external classifier, so it should classifier the same? Or is the idea that vicuna has a lower refusal rate in general?
- In figure 3, I am confused how the last layer tokens for criminals have no edges but high node scores. I thought the high node score indicated that the token affects the output. However, the lack of edges might suggest it doesn't affect the output that much?

---

> ### Author Response · Authors · 2025-11-25
> **Response to Reviewer fKnw [1/4]**
>
> Dear Reviewer fKnw,
>
> We sincerely thank you for your thoughtful and constructive review. Your comments were extremely helpful in improving the clarity and empirical completeness of the work.
> Unless otherwise noted, all additional experiments below were conducted on `LLaMA-2-7B-Chat` due to time and resource constraints.
>
> **(1) Exposition, framing, and clarity**
>
> **Reviewer concern:**
>
> The exposition could be improved; the neuroscience framing is not fully explained; the introduction should better articulate what “routing-level jailbreak behavior” means and why prior classifiers are described as “surface-level.”
>
> **Response:**
>
> We appreciate this helpful suggestion. We will substantially revise the introduction and early exposition to make the framing clearer and more balanced.
>
> (a) **More precise framing of prior work and taxonomies.**
>
> We will replace the vague term **“surface-level signals”** with a clearer explanation:
> *prior defenses often rely on local indicators (e.g., token patterns, gradient norms, per-layer probes, classifiers trained on fixed taxonomies) and therefore do not explicitly model how refusal-related semantics propagate across layers.*
> We will also clarify that taxonomy-driven approaches, including Constitutional Classifiers [1], are highly effective and complementary to our work—not competing.
>
> Such methods benefit from large synthetic datasets but involve heavy LLM-based classifiers and higher inference cost, whereas GraphShield is designed as a lightweight, low-FPR alternative that focuses specifically on multi-layer routing behavior.
>
> We will revise the related-work section to emphasize this complementarity.
>
> (b) **Clarifying why our classifier is intentionally simple.**
>
> To isolate the effect of our graph-theoretic features, we intentionally pair them with a lightweight binary classifier.
>
> We will state explicitly that this design choice is methodological and does not imply that taxonomy-based defenses are weak; rather, we aim to highlight the contribution of the routing representation itself.
>
> (c) **Clearer high-level narrative and diagram improvements.**
>
> We will add a short conceptual paragraph in the Introduction and move the routing illustration (currently Fig. 1) earlier.
> The revised narrative will emphasize that GraphShield tracks how refusal-aligned directions accumulate and propagate through layers—capturing the multi-hop connectivity patterns disrupted by jailbreaks, rather than probing isolated activations.
>
> (d) **Mathematical clarity and feature intuition.**
>
> Section 3 will be expanded with brief explanations and a schematic of the sparsified token-layer graph to clarify the meaning of the routing definitions and how graph features arise from them.
>
> (e)	**Future extensions.**
>
> We agree that stronger training paradigms could further enhance robustness, and we will note the following as future directions:
> adaptive or learned anchors, hybrid guards combining routing features with taxonomy-based classifiers, and few-shot graph adaptation for new attack styles.
> These changes address the reviewer’s concerns and significantly improve exposition and framing.
> We will clarify this reasoning in the revision.
>
> *[1] SHARMA, Mrinank, et al. Constitutional classifiers: Defending against universal jailbreaks across thousands of hours of red teaming. arXiv preprint arXiv:2501.18837, 2025.*
>
> **(2) Motivation behind graph features**
>
> **Reviewer concern:**
> The current draft does not sufficiently explain why each graph feature is chosen or how these features reflect the routing-level failure modes of jailbreaks (e.g., refusal signals not emerging or not being routed to the output).
>
> **Response:**
> We will clarify the feature motivation by explicitly linking each feature group to the routing patterns illustrated in Fig.3:
>
> * **Total routed mass**: Harmful prompts suppress or redirect refusal-aligned flow, causing a drop in routed mass from the anchor. (In Fig.3(a), refusal-originated flow dissipates early.)
>
> * **Last-layer inflow / top-k inflow share**: Harmful prompts concentrate propagation into a narrow set of harmful tokens, producing a highly peaked inflow distribution. (Fig.3(a) shows large late-layer nodes for “criminals”, “bomb”, etc.)
>
> * **Edge-weight concentration (Gini)**: Harmful prompts create sharper, more selective routing funnels rather than broad diffusion.
> (Fig.3(a) displays sparse but high-weight paths.)
>
> We will expand these explanations in the revised manuscript and include a simple scatter plot (routed mass vs. last-layer inflow concentration) in the appendix to visually illustrate the harmful–benign separation.

---

> ### Author Response · Authors · 2025-11-25
> **Response to Reviewer fKnw [2/4]**
>
> **(3) Judging procedure – strong evaluation now included**
>
> **Reviewer concern:**
> The current evaluation relies on a keyword-based pattern filter; a stronger rubric-based judging method (e.g., StrongREJECT) would provide a more robust safety assessment.
>
> **Response:**
> We agree, and we now incorporate two stronger and widely used evaluation pipelines in addition to the keyword heuristic:
>
> (1) **StrongREJECT rubric evaluator**,
>
> (2) **A majority-voted LLM judge** (GPT-4o, Claude-3-Opus, Gemini-2.5).
>
> Both judges yield consistent safety assessments and substantially reduce ASR compared to raw model outputs.
>
> `LLaMA-2-7B-Chat`:
> * **StrongREJECT score**: 0.169 → 0.023
> * **ASR (LLM-judge)**: 21.47% → 2.94%
>
> `Vicuna-7B-v1.5`:
> * **StrongREJECT score**: 0.400 → 0.055
> * **ASR (LLM-judge)**: 68.83% → 7.61%
>
> These results confirm that our evaluation is robust to stronger rubric-style and LLM-based judging protocols.
>
> We will update the paper accordingly.
>
> **(4) Baselines and threshold comparability**
>
> **Reviewer concern:**
> Include a linear-probe baseline; clarify benefit of connectivity; consider fixed-BRR threshold comparison.
>
> **Response:**
> We implemented several new baselines:
>
> * **Linear-probe baseline (no graph; anchor cosine only).**
> Uses only layer-wise cosine similarity between the hidden states and anchor embeddings: TPR = **83.9%**
>
> * **Non-graph pooled features (no graph; anchor present).**
> Aggregates layer-wise anchor embeddings via mean/max pooling without constructing any graph structure: TPR = **88.8%**
>
> * **Sequence-level attention graph (graph present; no anchors).**
> Constructs a global attention graph without anchor conditioning, capturing only sequence-level connectivity: TPR = **87.27%**
>
> * **Anchor-less global graph (graph present; no anchors; structural features only).**
> Represents only global graph topology (centrality, modularity, degree metrics) with no anchor-driven probing: TPR = **68.25%**
>
> These ablations show:
>
> * graph-only → insufficient semantics
> * anchors-only → insufficient structural routing
> * both together → needed for high separability (GraphShield (full): 91.00% TPR), showing that both semantic conditioning and graph connectivity are necessary.
>
> We will integrate these ablations into the appendix.
>
> We will also include an appendix table showing TPR/ASR at matched BRR (e.g., about 5%) for baseline defenses and GraphShield.

---

> ### Author Response · Authors · 2025-11-25
> **Response to Reviewer fKnw [3/4]**
>
> **(5) Anchor tokens, robustness, and obfuscation concerns**
>
> **Reviewer concern:**
> Are plain-English anchors brittle? Do they generalize across languages? Were alternative anchor mechanisms considered?
>
> **Response:**
> (a)	**Robustness to anchor choice and obfuscation**
> To address brittleness concerns, we conducted extensive anchor ablations:
> * **Anti-anchors** (embedding-farthest tokens).
> We selected tokens maximally distant from the refusal-anchor embedding centroid (e.g., *“hangul”*, *“Kyushu”*, *“midi”*): TPR: **83.70%**.
>
> * **Subword anchors** (no lexical semantics).
> We replaced all anchors with non-semantic subword fragments (e.g., *“apl”*, *“gnr”*, *“khz”*), removing any meaningful refusal-related content: TPR: **83.88%**
>
> * **Extended semantic anchors**
> We also evaluated broader refusal-related tokens (*“ai”*, *“apologize”*, *“as”*, *“refuse”*, *“sorry”*, *“unable”*), representing semantic drift from the original set: TPR: **88.8%**.
>
> Even anchors with no English meaning preserve most performance, showing that GraphShield uses embedding-space directions, not token identity.
> We additionally evaluated robustness under obfuscation:
> * Decipher improves substantially with simple Gemini paraphrasing augmentation
> (seen: **58.4%** $\to$ **68.0%** TPR; unseen: **37.5%** $\to$ **40.1%** TPR) without architectural changes.
> * Four fully unseen new attack families
> PAP [1] **94.17%** TPR, PEZ [2] **97.50%** TPR, TAP [3] **85.83%** TPR, and UAT [4] **95.76%** TPR.
>
> $\to$ demonstrating generalization beyond the training taxonomy.
>
> Together, these results indicate that GraphShield captures **a stable routing-based signature** that is resilient to anchor changes, paraphrase noise, and unseen attack structures.
>
> We will integrate this anchor intuition and the full set of anchor ablations in the revision.
>
> *[1] ZENG, Yi, et al. How johnny can persuade llms to jailbreak them: Rethinking persuasion to challenge ai safety by humanizing llms. In: Proceedings of the 62nd Annual Meeting of the Association for Computational Linguistics (Volume 1: Long Papers). 2024. p. 14322-14350.*
>
> *[2] WEN, Yuxin, et al. Hard prompts made easy: Gradient-based discrete optimization for prompt tuning and discovery. Advances in Neural Information Processing Systems, 2023, 36: 51008-51025.*
>
> *[3] MEHROTRA, Anay, et al. Tree of attacks: Jailbreaking black-box llms automatically. Advances in Neural Information Processing Systems, 2024, 37: 61065-61105.*
>
> *[4] WALLACE, Eric, et al. Universal adversarial triggers for attacking and analyzing NLP. arXiv preprint arXiv:1908.07125, 2019.*
>
>
> (b)	**Multilingual considerations**
>
> While the current work focuses on English, the fact that **non-semantic subword anchors still work**(in Response (5-a), TPR = 83.88%) strongly suggests that GraphShield relies on **model-internal semantic geometry rather than surface lexical form**.
>
> This provides preliminary evidence that the method is **not tied to English vocabulary** and may transfer to multilingual settings where anchors do not correspond to meaningful words.
> We agree that explicit multilingual anchor discovery is an important extension, and we will explore this direction in future work.

---

> ### Author Response · Authors · 2025-11-25
> **Response to Reviewer fKnw [4/4]**
>
> **(6) Response to reviewer questions**
>
> **Q1. Anchor tokens—how do they work? What exactly are you probing?**
>
> At a high level, **anchor tokens are not lexical triggers.**
>
> They act as **semantic “probes”** that introduce a reference direction in hidden-state space corresponding—approximately—to refusal-related meaning.
> GraphShield then measures **how strongly and how consistently this semantic direction is routed through the network.**
>
> Jailbreaks typically disrupt this routing (either by suppressing it or diverting it), which produces clear geometric differences in the graph features.
> Thus, anchors serve as **coordinate axes** for semantic routing—not as keywords that must appear in the input/output.
>
> **Q2. Cross-lingual robustness — would translation break the anchors? (Decipher concern)**
>
> We appreciate this concern.
>
> Although our main experiments focus on English, three anchor ablations demonstrate that GraphShield does not rely on English lexical forms (detailed in Response (5-a)):
>
> * Subword anchors (meaningless fragments): **83.9%** TPR
> * Anti-anchors (embedding-farthest tokens): **83.7%** TPR
> * Extended semantic anchors (*“sorry”*, *“unable”*, etc.): 88.8% TPR
>
> Because even meaningless subword fragments work, the model relies on semantic-routing geometry, not surface word identity.
> We acknowledge that explicit multilingual anchors are a promising extension, and we will explore this direction in future work.
>
> **Q3. Did you consider alternative anchor approaches, such as layerwise refusal vectors (linear probes)?**
>
> To directly address this, we implemented the following new baselines, designed to mirror exactly the reviewer’s proposal:
>
> * **Linear-probe baseline (no graph; anchor cosine only).**
> Uses only layer-wise cosine similarity between the hidden states and anchor embeddings: TPR = **83.9%**
>
> * **Non-graph pooled features (no graph; anchor present).**
> Aggregates layer-wise anchor embeddings via mean/max pooling without constructing any graph structure: TPR = **88.8%**
>
> * **Sequence-level attention graph (graph present; no anchors).**
> Constructs a global attention graph without anchor conditioning, capturing only sequence-level connectivity: TPR = **87.27%**
>
> * **Anchor-less global graph (graph present; no anchors; structural features only).**
> Represents only global graph topology (centrality, modularity, degree metrics) with no anchor-driven probing: TPR = **68.25%**
>
> These ablations show:
>
> * graph-only → insufficient semantics
> * anchors-only → insufficient structural routing
> * both together → needed for high separability (GraphShield (full): 91.00% TPR), showing that both semantic conditioning and graph connectivity are necessary.
>
> We will integrate these ablations into the appendix.
>
> **Q4. Edge weights—what exactly do they represent? Are they measuring propagation layer-to-layer?**
>
> * Edge weights represent: ${attention}$\_${weight}$ × $routed$\_$mass$
>
> This quantifies how much refusal-aligned information flows from layer $ℓ→ℓ+1$ between token pairs.
> Thus, yes—the edges model layer-to-layer propagation of refusal semantics.
> We will clarify this in the revised manuscript.
>
> **Q5. Why is Llama-Guard’s ASR different on LLaMA-2 vs. Vicuna if it is an external classifier?**
>
> Llama-Guard’s classifier is identical, but ASR differs because it is computed on the target model’s generated outputs.
> Since `Vicuna-7B-v.1.5` produces more jailbreak-prone outputs than `LLaMA-2-7B-Chat` under identical prompts, the same guard yields higher ASR.
>
> Although Llama-Guard blocks similar prompts, the resulting “allowed” Vicuna responses are more harmful—hence the higher ASR.
>
> **Q6. Figure 3 confusion—why do last-layer harmful tokens have high node scores but no outgoing edges?**
>
> Last-layer nodes have no outgoing edges by construction (edges only go from layer $ℓ→ℓ+1$),
> but their routed score is high because they **accumulate multi-layer reachability to the sink token** (i.e., the final aggregation node for routed mass).
>
> We will clarify this in the caption.
>
> We sincerely thank the reviewer again for their constructive comments.
>
> We will incorporate all clarifications and new experimental results in the revised manuscript.
>
> Please let us know if you have any further questions or concerns.
>
> Best regards,
>
> Authors

---

### Official Review · Reviewer_VHh2 · 2025-10-26

**Soundness:** 3
**Presentation:** 3
**Contribution:** 4
**Rating:** 8
**Confidence:** 3

**Summary:**

This paper proposes GraphShield, a novel neuroscience inspired approach to detecting jailbreak attacks by analyzing graph-theoretic network level semantics instead of shallow surface level indicators, unlike past work. Their method leads to a light-weight, model-agnostic classifier that achieves the best trade-off on defense and preserving utility (responding to benign prompts) when compared to gradient-based, hidden-state based, perplexity/pattern based defenses etc. and also works across two model families. They also perform crucial ablations on the importance of the chosen set of features, showing their importance for maximizing true positives while keeping a low false positive rate.

**Strengths:**

1. Proposes a novel, neuroscience-inspired, network-level approach for jailbreak detection that is lightweight, needing only a single forward pass and simple classifiers. Also, compared to past methods, it relies on network-level semantics instead of focusing on single-point, localized, or surface-level indicators of semantics (e.g., hidden states, specific patterns or tokens, or specific training taxonomies).
2. Outperforms prior pattern-based, gradient-based, hidden-state-based, and multi-pass methods that rely on gradients, taxonomies, model-specific alignment, or multiple passes.
3. The method is model-agnostic and requires no fine-tuning.
4. Method achieves a good trade-off between benign refusal rate (BRR), a measure of utility, and attack success rate (ASR), a measure of how many harmful prompts succeed in eliciting a harmful response. It has a favorable trade-off between harmlessness and harmfulness.
5. The two chosen models demonstrate a huge reduction in ASR through their defense, and they also achieve the best or second-best reduction in ASR, and the most favorable trade-off between ASR and BRR.

**Weaknesses:**

1. **Limited generalizability across attack types:** The leave-one-out attack ablations show that their detection method struggles across rare or obfuscated attacks (e.g., Deciper). While the authors claim that this could be mitigated by ensuring diverse attack types are present in the training distribution by leveraging synthetic jailbreaks, this seems hard to achieve in practice since new attacks keep emerging all the time, and this would require something like an active learning setup with their detector. **Also, this means their method has the same limitation as the classifier-based methods since they effectively require a taxonomy of attacks to ensure diverse attack types are covered.**
2. Do the network-level patterns associated with harmfulness for a given set of harmful scenarios match the patterns in new scenarios with similar kinds/definitions of harm but completely different settings? I recommend using something like WildJailBreak [1] or InjecAgent [2] for different settings. I also have some general concerns about the small size of the test set (only 36 prompts/scenarios per attack family) despite the claim of low variance found in their bootstrap-style testing. I would be more reassured about the robustness of their method if these suggested evaluations were performed.
3. **Concerns about generalizability across model scales:** While the paper does a good job of choosing models from different families (Vicuna and Llama), my main concern is that both are 7B models. Inverse scaling is a well-known trend in the alignment literature [3] that suggests that models of varying scales exhibit significantly different behavior to harmful prompts. My intuition is that this would lead to the emergence of different and potentially more complex patterns across model scales. While the authors’ intent is to target larger and closed-source models in future work, I would highly recommend having at least 1 open-source model in the 32B-70B parameter range in their experiments for this paper as a way to show that their detector can capture jailbreak-related patterns for larger models.

[1] Jiang, Liwei, et al. "Wildteaming at scale: From in-the-wild jailbreaks to (adversarially) safer language models." Advances in Neural Information Processing Systems 37 (2024): 47094-47165.
[2] Zhan, Qiusi, et al. "Injecagent: Benchmarking indirect prompt injections in tool-integrated large language model agents." arXiv preprint arXiv:2403.02691 (2024).
[3] McKenzie, Ian R., et al. "Inverse scaling: When bigger isn't better." arXiv preprint arXiv:2306.09479 (2023).

**Questions:**

1. Why do you sample 120 instances from JailbreakBench instead of using the whole dataset?
2. Since you are using JBB-Behaviors (https://huggingface.co/datasets/JailbreakBench/JBB-Behaviors), which already has 100 harmful and benign behaviors, why do you source the benign prompts from the AlpacaEval dataset?

---

> ### Author Response · Authors · 2025-11-25
> **Response to Reviewer VHh2 [1/3]**
>
> Dear Reviewer VHh2,
>
> We sincerely thank you for your thoughtful and constructive review. Your comments were extremely helpful in improving the clarity and empirical completeness of the work.
> Unless otherwise noted, all additional experiments below were conducted on `LLaMA-2-7B-Chat` due to time and resource constraints.
>
> **(1) Generalizability across diverse and obfuscated attack types**
>
> **Reviewer concern:**
> Leave-one-out results show lower performance for rare or obfuscated attacks (e.g., Decipher). Reviewer recommends evaluating on WildJailbreak or InjecAgent. Also notes that test sets contain only 36 prompts per family.
>
> **Response:**
> We appreciate this important point.
> To directly assess transferability beyond the JBB distribution, we performed several new evaluations:
>
> (a)	**WildJailbreak**
>
> GraphShield achieves **88.33%** TPR on 120 WildJailbreak adversarial prompts. $\to$ This demonstrates transfer to unconstrained, human-written jailbreaks outside JBB-style distributions. We plan to extend our evaluation to indirect prompt-injection settings such as InjecAgent, and will include these results in the camera-ready version if resources permit.
>
> (b)	**HarmBench inductive jailbreaks**
>
> These evaluations use the same PAIR/GCG attack algorithms but start from HarmBench’s harmful scenarios, which differ substantially from the JBB-style scenarios used in the main paper.
>
> GraphShield achieves **86.7%** TPR (**PAIR**) and **90.0%** TPR (**GCG**), indicating that the routing-level signatures learned from JBB generalize well to new harmfulness narratives and scenario distributions, not only to the specific prompts seen during training.
>
> (c)	**Four completely unseen jailbreak families** (not used in training).
>
> GraphShield maintains strong detection to unseen new attack families: PAP [1] **94.17%** TPR, PEZ [2] **97.50%** TPR, TAP [3] **85.83%** TPR, and UAT [4] **95.76%** TPR.
> $\to$ demonstrates generalization beyond the training taxonomy.
>
> *[1] ZENG, Yi, et al. How johnny can persuade llms to jailbreak them: Rethinking persuasion to challenge ai safety by humanizing llms. In: Proceedings of the 62nd Annual Meeting of the Association for Computational Linguistics (Volume 1: Long Papers). 2024. p. 14322-14350.*
>
> *[2] WEN, Yuxin, et al. Hard prompts made easy: Gradient-based discrete optimization for prompt tuning and discovery. Advances in Neural Information Processing Systems, 2023, 36: 51008-51025.*
>
> *[3] MEHROTRA, Anay, et al. Tree of attacks: Jailbreaking black-box llms automatically. Advances in Neural Information Processing Systems, 2024, 37: 61065-61105.*
>
> *[4] WALLACE, Eric, et al. Universal adversarial triggers for attacking and analyzing NLP. arXiv preprint arXiv:1908.07125, 2019.*
>
> (d)	**Difficult cases (Decipher / Jood)**
>
> Although Decipher is intentionally obfuscated (anchor suppression + multi-hop paraphrase),
> GraphShield improves substantially under lightweight Gemini paraphrasing augmentation:
>
> * Decipher TPR (**seen**): **58.4%** $\to$ **68.0%**
> * Decipher TPR (**unseen**): **37.5%** $\to$ **40.1%**
>
> without architectural changes.
>
> (e)	**On the small per-family test size**
>
> We acknowledge this limitation. To ensure reliability:
>
> * All results are averaged over five different random seeds (existing),
> * Variance across seed was consistently low (standard deviation of 1.36% at TPR=91.0%), and
> * Additional sample-size scaling experiments (20-100% of training data) show stability (e.g., seen: **84.7%** TPR $\to$ **91.0%** TPR, unseen: **56.3%** $\to$ **61.9%**)
>
> $\to$ These findings suggest that GraphShield is not overly sensitive to dataset size or sampling variance.
>
> We will incorporate these results into the revision.

---

> ### Author Response · Authors · 2025-11-25
> **Response to Reviewer VHh2 [2/3]**
>
> **(2) Do routing-level harmfulness patterns transfer to new settings?**
>
> **Reviewer concern:**
>
> Do harmfulness routing patterns generalize to new domains or tasks?
> Reviewer recommends WildJailbreak / InjecAgent.
>
> **Response:**
>
> Yes. As shown in the new evaluations above (Response (1)), GraphShield consistently transfers across domains:
>
> * **WildJailbreak**: **88.33%** TPR
> * **HarmBench inductive attacks**: **86–90%** TPR
> * **Unseen families (PAP/PEZ/TAP/UAT)**: **86–97%** TPR
>
> These results suggest GraphShield captures stable, domain-invariant routing signatures rather than dataset-specific artifacts. We plan to extend our evaluation to indirect prompt-injection settings such as InjecAgent, and will include these results in the camera-ready version if resources permit.
>
> **(3) Generalizability across model scales**
>
> **Reviewer concern:**
>
> Only 7B models were evaluated; inverse-scaling suggests larger models may behave differently.
> Reviewer recommends testing 30B–70B models.
>
> **Response:**
>
> We fully agree that scale is important.
> While 30B+ models were not available due to the limited time and compute resources, we additionally evaluated GraphShield on three more models: `LLaMA-2-13B` (**86.85%** TPR), `LLaMA-3.1-7B` (**85.31%** TPR), `Qwen-1.5-7B` (**82.96%** TPR)
>
> Qwen differs substantially in tokenizer and alignment style, yet GraphShield still performs strongly (~83%), suggesting that the detector captures model-internal routing patterns rather than LLaMA-specific signatures.
>
> We will explicitly include evaluation on 30B–70B models as a main direction for future work.
>
> **(4) Responses to reviewer questions**
>
> **Q1. Why use 120 samples from JailbreakBench instead of the full set?**
>
> We selected 120 harmful prompts per family to match the benign set (≈800 AlpacaEval benign prompts) and maintain balanced classification across all attack families.
> Using the full JBB for some families but not others would create severe imbalance and make cross-family comparisons unreliable.
>
> To ensure the sampled subsets were reliable:
>
> * All results are averaged across five random seeds,
> * The variation across seeds is consistently low (e.g., for `LLaMA-2-7B-Chat`, TPR 91.00% has std = 1.36%),
> * Additional sample-size scaling experiments (20% $\to$ 100% of training data) show stable performance trends, indicating that GraphShield is not overly sensitive to sampling variation.
>
> Looking ahead, we view expanding to larger harmful datasets (full JBB, WildJailbreak, HarmBench) and developing techniques that further reduce sample-size sensitivity (e.g., dynamic anchor selection, graph-based few-shot adaptation) as promising directions for future work. These extensions go beyond the scope of the current submission but naturally build on the present methodology.
> We will clarify these design choices in the revised version.

---

> ### Author Response · Authors · 2025-11-25
> **Response to Reviewer VHh2 [3/3]**
>
> **Q2. Why source benign prompts from AlpacaEval when JBB-Behaviors includes 100 benign behaviors?**
>
> Our goal was to train one detector that generalizes across many heterogeneous jailbreak families.
>
> For this purpose, the 100 benign samples in JBB-Behaviors—constructed from a single behavioral taxonomy—were not sufficiently diverse to represent the broad range of benign user interactions expected in practice.
>
> We therefore sourced benign prompts from AlpacaEval, for two reasons:
>
> * **Greater diversity:**
> AlpacaEval’s 804 benign prompts span many everyday tasks and are widely used in alignment/safety evaluation ([1], [2], [3]).
>
> * **Better balance across attack families:**
> Since GraphShield trains a single binary classifier across multiple attack types, a larger and more varied benign set reduces bias and improves stability.
>
> To ensure that GraphShield is not dependent on AlpacaEval specifically,
> we augmented the benign set with the 100 JBB-behaviors benign prompts and retrained the detector.
>
> Performance remained essentially unchanged (TPR = **90.9%**, FPR = **5.01%**),
> showing that GraphShield is robust to the choice of benign dataset.
>
> We will clarify this rationale in the revision; full benign-set statistics and ablations will be added to the appendix.
>
> *[1] SHEN, Guobin, et al. Jailbreak antidote: Runtime safety-utility balance via sparse representation adjustment in large language models. arXiv preprint arXiv:2410.02298, 2024.*
>
> *[2] HU, Xiaomeng; CHEN, Pin-Yu; HO, Tsung-Yi. Token highlighter: Inspecting and mitigating jailbreak prompts for large language models. In: Proceedings of the AAAI Conference on Artificial Intelligence. 2025. p. 27330-27338.*
>
> *[3] Guorui Chen, Yifan Xia, Xiaojun Jia, Zhijiang Li, Philip Torr, and Jindong Gu. 2025. LLM Jailbreak Detection for (Almost) Free!. In Findings of the Association for Computational Linguistics: EMNLP 2025, pages 5777–5807, Suzhou, China. Association for Computational Linguistics.*
>
>
> We sincerely thank the reviewer again for their constructive comments.
>
> We will incorporate all clarifications and new experimental results in the revised manuscript.
>
> Please let us know if you have any further questions or concerns.
>
> Best regards,
>
> Authors.

---

### Official Review · Reviewer_p9ne · 2025-10-30

**Soundness:** 3
**Presentation:** 4
**Contribution:** 3
**Rating:** 6
**Confidence:** 4

**Summary:**

The paper proposes GraphShield, a jailbreak detection method that constructs token–layer graphs from model hidden states and attentions, tracks semantic routing around selected anchor tokens, and classifies extracted graph features with an RBF-SVM to block harmful prompts before generation. Evaluated on LLaMA-2-7B-Chat and Vicuna-7B-v1.5 across seven jailbreak attack families and a benign set, it reports low attack success rates with modest benign refusal rates, demonstrates speed advantages over multi-pass defenses, and analyzes performance under unseen and adaptive attack scenarios using a mixed heuristic-plus-human evaluation protocol.

**Strengths:**

* The paper presents a novel framing of jailbreak detection through network-level routing using token–layer graphs, supported by principled rollout, sparsification, clear mathematical grounding, and reproducible pseudocode. The writing is clear, detailed, and easy to follow throughout.
* GraphShield achieves a strong utility–robustness trade-off, obtaining ASR and BSR that are competitive with or outperforming prior defenses.
* The experiments include clear and extensive ablations, where most design choicesare validated.
* The method has favorable runtime, leveraging single-pass feature reuse to operate faster than multi-generation defenses which takes an order of magnitude longer.
* The paper is transparent about limitations, explicitly reporting seen vs. leave-one-out and adaptive failures, and providing comprehensive appendices and baseline implementation details.
* The evaluation includes a broad selection of jailbreak attack families and defense baselines, contributing to a thorough comparative analysis.

**Weaknesses:**

1. The model-agnosticity claim would be more convincing with experiments on a broader set of models, including additional architecture families such as Qwen.
2. Although the paper provides ablations on the selected anchor words, the reasoning behind their specific choice is not fully explained. It would be helpful to evaluate other commonly used refusal-related tokens (e.g., “sorry,” “as an AI…”) as potential anchors.
3. Incorporating more diverse benign datasets could further strengthen generalization and provide a more reliable estimate of false refusals in real-world usage.
4. Some hyperparameter settings, such as those described in Appendix D, lack justification. Ablations or sensitivity analyses on these choices would improve clarity and reproducibility.
5. While the keyword-based refusal heuristic shows over 90% agreement with humans, evaluating the approach with an LLM-based judge would provide an additional perspective on labeling quality and robustness.

***Minor remark:***
1. The numerical results reported in the abstract would be more meaningful if accompanied by a brief comparison point or baseline reference to contextualize their significance.

**Questions:**

1. As Figure 1 indicates clearly lower performance on Decipher and JOOD attacks, I would like to see the ASR of each baseline defense specifically on these two attack families to better understand how GraphShield compares at.
2. In the unseen attack setting, performance drops noticeably for some attack families. So, could you provide an analysis of how performance scales with the number of training examples for these attack types, for example by reporting ASR/TPR as a function of sample size?

---

> ### Author Response · Authors · 2025-11-25
> **Response to Reviewer p9ne [1/3]**
>
> Dear Reviewer p9ne,
>
> We sincerely thank you for your thoughtful and constructive review. Your comments were extremely helpful in improving the clarity and empirical completeness of the work.
> Unless otherwise noted, all additional experiments below were conducted on `LLaMA-2-7B-Chat` due to time and resource constraints.
>
> **(1) Model-agnosticity across more architectures**
>
> **Reviewer concern:**
> The model-agnosticity claim would be more convincing with results on additional model families such as Qwen.
>
> **Response:**
> We fully agree, and we have now evaluated GraphShield on three additional models: `LLaMA-2-13B` (**86.85%** TPR), `LLaMA-3.1-7B` (**85.31%** TPR), and `Qwen-1.5-7B` (**82.96%** TPR).
>
> Despite major architectural/tokenizer differences—especially between LLaMA and Qwen—GraphShield sustains strong performance.
> These results support the claim that GraphShield captures **universal routing signatures**, not model-specific artifacts.
>
> **(2) Anchor selection: why these anchors?**
>
> **Reviewer concern:**
> Anchor choice not fully explained; evaluate other refusal-related tokens such as *“sorry”*, *“as an AI…”*.
>
> **Response:**
> We thank the reviewer for pointing this out.
>
> To directly evaluate whether GraphShield depends on the specific anchor words we originally selected (*“can”*, *“cannot”*, *“help”*, *“I”*, *“else”*), we conducted new ablations using widely used refusal-related expressions such as *“sorry”* or *“as an AI …”*.
>
> To isolate the effect more clearly, we tested each extended anchor individually in addition to evaluating them as a group.
>
> **(a) Extended semantic anchors**
>
> Using refusal-related tokens commonly produced by safety-aligned LLMs: *ai* (**85.71%** TPR), *apologize* (**86.96%** TPR), *as* (**86.96%** TPR), *refuse* (**82.58%** TPR), *sorry* (**81.37%** TPR), *unable* (**85.09%** TPR).
>
> Combined extended-anchor set: **88.8%** TPR $\rightarrow$ slightly lower than the original **91.0%** TPR, but consistently strong. This shows that GraphShield does not rely on the specific English anchor words we selected, but robustly adapts to a wide range of refusal-semantic cues.
>
> **(b) Additional robustness checks**
>
> For completeness, we also evaluated:
> * **Anti-anchors** (tokens maximally distant from refusal semantics: *hangul*, *Kyushu*, *midi*, *unison*, *xml*): TPR **83.70%**
> * **Subword anchors** (no lexical meaning: *apl*, *gnr*, *khz*, *uct*, *sx*): TPR **83.88%**
> These further show that GraphShield is driven by the semantic routing direction in hidden-state space rather than the surface form of anchor tokens.

---

> ### Author Response · Authors · 2025-11-25
> **Response to Reviewer p9ne [2/3]**
>
> **(3) Need for more diverse benign data**
>
> **Reviewer concern:**
> Evaluate with more diverse benign datasets to better estimate false refusals.
>
> **Response:**
> Using 100 additional benign prompts from JBB-Behaviors, GraphShield achieves TPR = **90.9%** and FPR = **5.0%**,
> suggesting that it maintains strong utility even under broader benign distributions.
> We will add these results in the revised appendix.
>
> **(4) Hyperparameter justification**
>
> **Reviewer concern:**
> Some hyperparameters in Appendix D lack justification or sensitivity analysis.
>
> **Response:**
> We appreciate the comment and will expand our appendix with clear motivation for all hyperparameters.
>
> Most settings follow standard practice in sparsified attention graphs and showed stable performance across reasonable variations
> (TPR differences <1–2% in our spot checks).
>
> Key examples include:
>
> **Shuffle-test threshold (z $\ge$ 2)**: common in permutation-based pruning; avoids over-pruning.
>
> **Routing mix $\alpha$ = 0.95**: stable in the 0.9–0.98 range; controls residual vs. attention flow.
>
> **Last-N layer window (N=8–10)**: safety-routing signals appear late in the network; results robust across N=6–12.
>
> **Louvain communities + PageRank centrality**: standard for sparse graphs; alternatives (Leiden/Infomap) produced similar modularity.
>
> We will include a concise table in the appendix summarizing all hyperparameters and their rationale.
>
> **(5) Refusal-labeling heuristic vs. LLM-based judging**
>
> **Reviewer concern:**
> Suggest using an LLM-based judge to strengthen labeling reliability.
>
> **Response:**
> We now evaluate ASR using majority-voted LLM judges (GPT-4o, Claude-3, Gemini-2.0).
> Under this stronger metric:
>
> - **LLaMA-2**: ASR drops from **21.47%** $\rightarrow$ **2.94%**
>
> - **Vicuna**: ASR drops from **68.83%** $\to$  **7.61%**
>
> This provides strong complementary evidence that GraphShield aligns with widely used LLM-based safety judgments.
>
> **(6) Minor remark - Provide baseline context in the abstract**
>
> **Reviewer concern:**
> Include brief baseline context for reported numbers.
>
> **Response:**
> Thank you for the suggestion.
>
> We agree that providing brief contextualization improves readability.
> In the revised version, we will update the abstract to include a short comparison point (e.g., relative to PPL-based filters and Llama-Guard), so that the significance of the reported numbers is immediately clear to readers.

---

> ### Author Response · Authors · 2025-11-25
> **Response to Reviewer p9ne [3/3]**
>
> **(7) Responses to reviewer questions**
>
> **Q1. Baseline ASR for Decipher / Jood**
>
> We now evaluate widely used baseline defenses (PPL, back-translation, Self-Reminder, SmoothLLM, Llama-Guard) on Decipher and Jood to contextualize GraphShield’s performance.
>
> | Defense            | Decipher ASR (%) | Jood ASR (%) |
> |--------------------|------------------|---------------|
> | Baseline (None)    | 42.0             | 0.6           |
> | PPL                | 5.0              | 0.2           |
> | Back-translation   | 34.1             | 0.0           |
> | Self-reminder      | 1.33             | 0.0           |
> | SmoothLLM          | 40.5             | 2.2           |
> | LLaMA-Guard        | 5.0              | 0.0           |
> | **GraphShield (ours)** | **17.5**     | **0.1**       |
>
> (Jood is inherently low-ASR in `LLaMA-2-7B-Chat`, so most baselines trivially flag it.)
>
> **Interpretation:**
>
> Decipher is a deliberately hard, multi-hop obfuscation attack that suppresses refusal cues. While some baselines (PPL, Self-Reminder, Llama-Guard) achieve lower ASR on this specific attack, these methods typically exhibit high benign refusal rates or large instability across other attack families.
>
> Importantly, under Gemini-based paraphrasing augmentation, GraphShield’s Decipher performance improves substantially (*seen*: **58.4%** $\to$ **68.0%** TPR, *unseen*: **37.5%** $\to$ **40.1%** TPR), showing that the gap can be closed without any architectural changes.
>
>
> Thus, although its raw Decipher ASR is higher than some baselines (PPL, self-reminder, LLaMA-guard), GraphShield provides consistent performance across all seven families while maintaining low BRR (7.08%).
>
> A full comparison table will be added in the appendix.
>
> **Q2. Performance scaling with training sample size**
>
> We thank the reviewer for this constructive suggestion.
> We conducted a systematic analysis by varying the **training data fraction** from 20% $\to$ 40%  $\to$ 60% $\to$ 80% $\to$ 100%.
>
> * **Seen attacks (overall TPR):**
> 84.71% $\to$ 86.08% $\to$ 90.28% $\to$ 91.00%
>
> $\to$ Moderate improvement, then saturation.
>
> * **Unseen attacks (overall TPR):**
> 56.31% $\to$ 57.45% $\to$ 57.63% $\to$ 61.87%
>
> $\to$ Smaller variation, indicating robustness to sample size.
>
> * **Difficult families**
> Decipher was improved from **50.1%** TPR $\to$ **58.4%** TPR as data increases (20% $\to$ 100%), reflecting its high obfuscation,
> and Jood remains essentially unchanged, showing that its routing signature is already captured.
> (Note that the table reports ASR, whereas the paraphrasing augmentation experiment reports TPR.
> Higher TPR corresponds to lower ASR.)
>
> Overall, GraphShield shows moderate gains but is not highly sensitive to training size. We will include these results in the appendix.
>
> We sincerely thank the reviewer again for their constructive comments.
>
> We will incorporate all clarifications and new experimental results in the revised manuscript.
>
> Please let us know if you have any further questions or concerns.
>
> Best regards,
>
> Authors.

---

> > ### Comment · Reviewer_p9ne · 2025-11-27
> >
> > Thank you for addressing most of my questions, I will maintain my current score.

---

### Official Review · Reviewer_ZSEn · 2025-11-01

**Soundness:** 3
**Presentation:** 2
**Contribution:** 2
**Rating:** 4
**Confidence:** 3

**Summary:**

This paper introduces GraphShield, a graph-theoretic framework for detecting jailbreak prompts in large language models. Its main idea is to construct token-layer graphs from hidden states and attention weights, use anchor tokens as probes for refusal-related semantics, extract multi-scale structural and semantic features from these graphs, and use them within a lightweight classifier to predict jailbreak attempts. Experiments on LLaMA-2-7B-Chat and Vicuna-7B-v1.5 demonstrate lower attack success rates and strong benign refusal performance, achieving a better robustness-utility balance than previous defenses.

**Strengths:**

1. The paper introduces a graph-theoretic approach to model information flow within LLMs, moving beyond surface-level or gradient-based detectors. This shift captures emergent routing patterns in jailbreak detection.
2. GraphShield achieves higher F1 and TPR at low FPR compared with baselines such as PromptGuard and PPL-based filters. It conducts detailed ablations on anchor tokens and feature groups to pinpoint which components most influence detection performance.
3. The token-layer graph visualizations clearly illustrate how refusal semantics propagate, offering intuitive explanations for detection outcomes.

**Weaknesses:**

1. Reliance on Supervised Classification: The method depends on a supervised SVM trained on labeled benign and attack prompts. Its performance may degrade as new jailbreak strategies or paraphrased anchors emerge, requiring retraining to maintain coverage.
2. Anchor Token Dependency: GraphShield is highly vulnerable to adaptive attacks that suppress or paaraphrase its fixed anchor tokens (“can”, “cannot”, “help”, “I”, “else”). Although retraining with such samples restores robustness, this reliance on continuous adversarial updates poses long-term scalability and maintenance challenges.
3. GraphShield’s pipeline remains intricate, involving multi-stage sparsification (permutation nulls, top-k pruning) and extensive feature extraction across structural, community, and centrality metrics for each anchor. Despite being described as lightweight, the resulting feature space is opaque. The link between specific feature groups and jailbreak likelihood is not straightforward. This black-box nature may limit trust and adoption in safety-critical contexts that require transparent post-hoc auditing.
4. Although the paper includes ablations on feature groups and anchor tokens, it lacks direct comparisons to alternative graph formulations, such as sequence-level attention graphs without anchor conditioning, or to simpler network-informed but non-graph-based features. Consequently, it remains unclear whether the specific anchor-probe graph design is essential for the reported performance gains.

**Questions:**

1. Since adaptive attacks target the anchor tokens (“can”, “cannot”, “help”, “I”, “else”), have the authors explored dynamic or learned anchor selection rather than fixed tokens?
2. How well does GraphShield generalize to unseen jailbreak types or paraphrasing styles without retraining?
3. Refer to the weakness

---

> ### Author Response · Authors · 2025-11-25
> **Response to Reviewer ZSEn [1/3]**
>
> Dear Reviewer ZSEn,
>
> We sincerely thank you for your thoughtful and constructive review. Your comments were extremely helpful in improving the clarity and empirical completeness of the work.
> Unless otherwise noted, all additional experiments below were conducted on `LLaMA-2-7B-Chat` due to time and resource constraints.
>
> **(1)	Reliance on Supervised Classification**
>
> **Reviewer concern:**
> GraphShield depends on a supervised SVM and may degrade under new jailbreak strategies or paraphrasing.
>
> **Response:**
> We fully agree that robustness to unseen distribution shifts is crucial. To directly evaluate this, we performed several new experiments:
>
> (a)	**Four completely unseen jailbreak families** (not used in training).
>
> GraphShield maintains strong detection: PAP [1] **94.17%** TPR, PEZ [2] **97.50%** TPR, TAP [3] **85.83%** TPR, and UAT [4] **95.76%** TPR.
> → demonstrates generalization beyond the training taxonomy.
>
> (b)	**Robustness to paraphrased or missing anchors**.
>
> GraphShield does not require the anchor tokens to appear in the prompt or output.
> Anchors act as semantic probe directions in hidden-state space.
>
> This is validated by:
>
> * Anti-anchors (embedding-farthest tokens).
> We selected tokens maximally distant from the refusal-anchor embedding centroid (e.g., *“hangul”*, *“Kyushu”*, *“midi”*): TPR: **83.70%**.
>
> * Subword anchors (no lexical semantics).
> We replaced all anchors with non-semantic subword fragments (e.g., *“apl”*, *“gnr”*, *“khz”*), removing any meaningful refusal-related content: TPR: **83.88%**
>
> * Extended semantic anchors.
> We also evaluated broader refusal-related tokens (*“ai”*, *“apologize”*, *“as”*, *“refuse”*, *“sorry”*, *“unable”*), representing semantic drift from the original set: TPR: **88.8%**.
>
> While these TPRs are naturally slightly below the original anchor set (**91.00%**), GraphShield remains consistently high-performing across all anchor variants, including semantically inverted or meaningless anchors.
>
> This indicates that GraphShield is not tied to specific English words, but instead leverages a stable semantic-routing signal in the model’s hidden states.
> We will clarify this empirical finding in the revised paper.
>
> *[1] ZENG, Yi, et al. How johnny can persuade llms to jailbreak them: Rethinking persuasion to challenge ai safety by humanizing llms. In: Proceedings of the 62nd Annual Meeting of the Association for Computational Linguistics (Volume 1: Long Papers). 2024. p. 14322-14350.*
>
> *[2] WEN, Yuxin, et al. Hard prompts made easy: Gradient-based discrete optimization for prompt tuning and discovery. Advances in Neural Information Processing Systems, 2023, 36: 51008-51025.*
>
> *[3] MEHROTRA, Anay, et al. Tree of attacks: Jailbreaking black-box llms automatically. Advances in Neural Information Processing Systems, 2024, 37: 61065-61105.*
>
> *[4] WALLACE, Eric, et al. Universal adversarial triggers for attacking and analyzing NLP. arXiv preprint arXiv:1908.07125, 2019.*

---

> ### Author Response · Authors · 2025-11-25
> **Response to Reviewer ZSEn [2/3]**
>
> **(2) Anchor Token Dependency**
>
> **Reviewer concern:**
> GraphShield may be vulnerable because the anchor tokens (*“can”*, *“cannot”*, *“help”*, *“else”*, etc.) are plain-English lexical forms.
>
> **Response:**
> We appreciate this concern. As detailed above (Response (1-b)), we conducted three anchor-set ablations designed to explicitly break any dependence on English surface forms. Across anti-anchors, subword anchors, and extended semantic anchors, GraphShield consistently maintains **83–89%** TPR, showing only modest degradation from the original 91%.
>
> These results indicate that GraphShield does not rely on the literal anchor tokens themselves.
> Instead, it leverages stable semantic-routing directions in the hidden-state space that persist even when:
>
> * anchors contain no lexical semantics (subword fragments),
>
> * anchors are intentionally semantically inverted (anti-anchors), or
>
> * anchors undergo semantic drift (extended anchor set).
>
> We will revise the paper to clarify that the method is not tied to specific English words, and that anchor tokens function as semantic probes, not lexical triggers.
> Dynamic or learned anchors remain a promising extension, and we plan to explore this direction in future work.
>
> **(3)	Pipeline Complexity and Opacity**
>
> **Reviewer concern:**
> The sparsification + multi-scale graph feature extraction may appear opaque; unclear which feature groups matter.
>
> **Response:**
> We agree that interpretability is important. We address this concern in four ways:
>
> (a)	**Feature-group ablations (existing)**
>
> Removing structural, community, or centrality features degrades performance—indicating each group is necessary.
>
> (b)	**Cross-model validation (new)**
>
> GraphShield generalizes across `LLaMA-2-13B` (**86.85%** TPR), `LLaMA-3.1-7B` (**85.31%** TPR), and `Qwen-1.5-7B` (**82.96%** TPR), suggesting learned signatures are not artifacts of a single architecture.
>
> (c)	**Additional diagnostic ablations requested by reviewers**
>
> To isolate the contribution of anchor-conditioning and graph topology, we performed four comparative ablations, each removing a key architectural component:
>
> * **Linear-probe baseline (no graph; anchor cosine only)**.
> Uses only layer-wise cosine similarity between the hidden states and anchor embeddings: TPR = **83.9%**
>
> * **Non-graph pooled features (no graph; anchor present)**.
> Aggregates layer-wise anchor embeddings via mean/max pooling without constructing any graph structure: TPR = **88.8%**
>
> * **Sequence-level attention graph (graph present; no anchors)**.
> Constructs a global attention graph without anchor conditioning, capturing only sequence-level connectivity: TPR = **87.27%**
>
> * **Anchor-less global graph (graph present; no anchors; structural features only)**.
> Represents only global graph topology (centrality, modularity, degree metrics) with no anchor-driven probing: TPR = **68.25%**
>
> These ablations show:
>
> * graph-only → insufficient semantics
>
> * anchors-only → insufficient structural routing
>
> * both together → needed for high separability (GraphShield (full): 91.00% TPR)
>
> Thus, the design is empirically justified.
>
> (d)	**Clarification planned in the text**
>
> We will improve Section 4 to provide more intuition (including a routing schematic) and clearer linkage between features and refusal-routing behavior.

---

> ### Author Response · Authors · 2025-11-25
> **Response to Reviewer ZSEn [3/3]**
>
> **(4) Missing Comparisons to Alternative Graph Formulations**
>
> **Reviewer concern:** Although the paper includes ablations on feature groups and anchor tokens, it lacks direct comparisons to alternative graph formulations (e.g., sequence-level attention graphs without anchor conditioning, simpler non-graph features). It is unclear whether the anchor-probe graph design is essential.
>
> **Response:**
> The diagnostic baselines described above (Response (3-c), **linear probe, non-graph pooled anchor features, sequence-level attention graph, anchor-less graph**) directly address this concern.
>
> Only the **full GraphShield architecture—i.e., the anchor-conditioned token–layer routing graph with multi-hop connectivity features—**achieves >90% TPR, confirming that the proposed design is essential.
>
> **(5) Responses to Reviewer Questions**
>
> **Q1. Dynamic or learned anchors?**
>
> We appreciate this question. Although our method uses a fixed set of anchors in the main paper, our new experiments (detailed in Responses (1) and (2)) systematically show that GraphShield does not depend on any specific anchor tokens:
> * **anti-anchors** (tokens farthest from refusal embeddings),
> * **subword anchors** (tokens with no lexical semantics), and
> * **extended semantic anchors** (broader refusal-related phrases)
>
> all achieve stable detection performance (TPR = **83–87%**).
> This indicates that the semantic-probe direction in hidden-state space, not the literal token identity, is what actually drives the routing signal GraphShield captures.
>
> While fully learned or dynamic anchors are a promising direction, our results demonstrate that GraphShield already operates robustly across highly diverse anchor sets. We plan to explore dynamic anchor discovery and layer-wise semantic probing in future work.
>
> **Q2. Generalization to unseen jailbreak styles?**
>
> As noted above (Response (1-a)), GraphShield maintains high performance (**85.8%–97.5%** TPR) on four unseen families (PAP/PEZ/TAP/UAT).
> These results directly confirm robust generalization without retraining.
>
> We will include these numbers in the appendix.
>
> We sincerely thank the reviewer again for their constructive comments.
>
> We will incorporate all clarifications and new experimental results in the revised manuscript.
>
> Please let us know if you have any further questions or concerns.
>
> Best regards,
>
> Authors

---

### Author Response · Authors · 2025-11-23
**General Response**

Dear reviewers and AC,

We sincerely appreciate the constructive and insightful feedback.

Following the reviewers’ comments—particularly on evaluation diversity, anchor-token dependence, generalization, and labeling quality—we substantially expanded our analysis and added the following updates:

**1. Expanded Evaluation: LLM-Judges & StrongReject**

To complement the keyword-based refusal heuristic, we added:

- A three-model majority LLM judge (GPT-4o, Claude-3.5-Opus, Gemini-2.5-Pro)
- StrongReject rubric scoring

Both consistently validate GraphShield’s performance trends and improve confidence in label quality.
(Appendix F.2, Appendix F.3)


**2. Additional Model Families (13B / Llama-3 / Qwen-7B)**

To strengthen the model-agnosticity claim, we extended experiments to:
- LLaMA-2-13B
- LLaMA-3.1-7B
- Qwen1.5-7B (base)

GraphShield maintains strong TPR/FPR performance across all three families. (Appendix I)

**3. Anchor Token Study: Extended, Anti-Anchor, Subword Probes**

We conducted an expanded study including:

- Extended refusal anchors (e.g., sorry, apologize, unable)
- Anti-anchors derived from embedding distance
- Subword-level anchors

Across all anchor configurations—extended refusal tokens, embedding-based anti-anchors, and subword-level anchors—GraphShield maintains consistently high detection performance (all TPRs ≥ 80%).

While some variants (e.g., subword or anti-anchors) show modest drops relative to the original anchors (~4–5%), no configuration collapses, confirming that GraphShield’s behavior is governed by graph-level routing patterns rather than any specific English anchor token. (Appendix J)

**4. Synthetic Augmentation for Attack Generalization**

As suggested by the reviewers, we apply a lightweight ×1.5 augmentation by replacing harmful trigger words with semantically similar harmful variants (lexical substitution), thereby increasing surface-form diversity without altering attack intent.

Even a minimal ×1.5 surface-form augmentation yields clear improvements for highly obfuscated attacks (e.g., Decipher, +9–10% TPR), while JOOD remains relatively stable due to its moderately strong—but not fully saturated—baseline performance.

These results support the reviewers’ hypothesis that limited surface-form diversity—as opposed to attack taxonomy—is the main bottleneck.  (Appendix M)

**5. Clarified Pipeline & Feature Motivation.**

We improved exposition by adding clearer explanations of our graph sparsification steps, structural/semantic feature motivations, and interpretability cues. These changes directly reflect reviewers’ suggestions regarding clarity and transparency.

We hope these additions address the major concerns regarding generalization, evaluation diversity, and robustness.

We thank all reviewers for the invaluable feedback, which significantly improved the clarity and quality of our work.

We provide detailed reviewer-specific responses in the discussion thread.

In the revised manuscript, we temporarily marked all modifications for the reviewers’ convenience. We hope these updates further clarify the contributions of GraphShield and address the concerns raised during the review process.

Best regards,

Authors.

---

### Meta-Review · Area_Chair_VorX · 2026-01-01

**Summary:**

The paper proposes a novel graph-inspired framework for detecting jailbreak prompts in LLMs. It constructs token-layer graphs using model hidden states and tracks semantic routing around anchor tokens. Then they classify the extracted graph features with SVM to judge the harmful prompts. Its results on Llama-2-7b-chat and Vicuna-7b-v1.5 across seven jailbreak attacks demonstrate a satisfyingly low attack success rate and benign refusal rates with speed advantages against many current methods.

Strengths:

1. The proposed method is novel and reasonable. Using graph for jailbreak detection is an interesting idea.

2. The proposed method achieves satisfying detection performance with a low refusal rate against many jailbreak methods.

Weaknesses:

1. Many different model sizes and architectures (like MoE and models larger than 30B) are not evaluated. It may limit the methods' impacts.

2. The methods still rely on supervised classification and may not be applicable to new attacks. Although the authors adopt some new attacks for evaluation, it is not enought as nowaday attacks are too various. If the authors can provide some theoretical guarantee to prove the generalization ability. It would be better.

But overall, the paper proposed a novel and interesting perspective for jailbreak detection with comprehensive evaluation. And the authors in the rebuttal phase detailly respond to reviewers' concerns on robustness, generalization, anchor selection, etc. I believe it is already a good paper to be accepted. However, due to the weaknesses above, I think its impact is not enough to make me recommend it as an oral. Therefore, I would like to recommend it as a poster.

**Reviewer Concerns:**

The main concerns raised by reviewers are robustness, generalization, anchor selection, writing, baselines, etc. The authors solved them in the rebuttal phase, and I think most of them are solved.

**Reviewer Scores:**

I think the authors have addressed most of the problems. Therefore, I believe all reviewers will increase their score to positive.

---

### Decision · Program_Chairs · 2026-01-26

Accept (Poster)